# Structural, Optical and Electrical Characterizations of Midwave Infrared Ga-Free Type-II InAs/InAsSb Superlattice Barrier Photodetector †

**U. Zavala-Moran [1,2], M. Bouschet [2,3], J. P. Perez [2], R. Alchaar [2], S. Bernhardt [4], I. Ribet-Mohamed [4], F. de Anda-Salazar [1] and P. Christol [2,*]**

[1] IICO, Univ. Autónoma de San Luis Potosí, Av. Karakorum 1470, San Luis Potosí CP 78210, Mexico; ulises.zavala-moran@ies.univ-montp2.fr (U.Z.-M.); francisco.deanda@uaslp.mx (F.d.A.-S.)

[2] IES, Univ. Montpellier, CNRS, F-34000 Montpellier, France; maxime.bouschet@ies.univ-montp2.fr (M.B.); perez@ies.univ-montp2.fr (J.P.P.); alchaar@ies.univ-montp2.fr (R.A.)

[3] LYNRED, BP 21, 38113 Veurey-Voroize, France

[4] ONERA, Chemin de la Hunière, F-91761 Palaiseau, France; sylvie.bernhardt@onera.fr (S.B.); isabelle.ribet@onera.fr (I.R.-M.)

[*] Correspondence: christol@ies.univ-montp2.fr

[†] This paper is an extended version from our paper, U. Zavala-Moran; R. Alchaar; J. P. Perez; J. B. Rodriguez; M. Bouschet; V. H. Compean; F. de Anda; P. Christol Antimonide-based Superlattice Infrared Barrier Photodetectors. In Proceedings of the 8th International Conference on Photonics, Optics and Laser Technology (PHOTOPTICS 2020), pages 45–51; https://doi.org/10.5220/0009004900450051.

**Abstract:** In this paper, a full set of structural, optical and electrical characterizations performed on midwave infrared barrier detectors based on a Ga-free InAs/InAsSb type-II superlattice, grown by molecular beam epitaxy (MBE) on a GaSb substrate, are reported and analyzed. a Minority carrier lifetime value equal to 1 μs at 80 K, carried out on dedicated structure showing photoluminescence peak position at 4.9 μm, is extracted from a time resolved photoluminescence measurement. Dark current density as low as $3.2 \times 10^{-5}$ A/cm$^2$ at 150 K is reported on the corresponding device exhibiting a 50% cut-off wavelength around 5 μm. A performance analysis through normalized spectral response and dark current density-voltage characteristics was performed to determine both the operating bias and the different dark current regimes.

**Keywords:** midwave infrared quantum detector; barrier structure; ga-free type-II superlattice

## 1. Introduction

Today, high performance, high speed cooled photodetectors operating in the midwave infrared (MWIR) spectral domain between 3 μm and 5 μm are of great interest for specific applications such as cancer diagnosis, gas analysis, astronomy, search and rescue in harsh environments and night vision.

The maximum operating temperature of a semiconductor IR photodetector is usually determined by its dark current, which increases exponentially with temperature. Therefore, in order to maintain a high signal-to-noise ratio (or a low dark current value) of the focal plane arrays (FPAs) of IR photodetectors, it is necessary to reduce the operating temperature down to cryogenic temperatures (typically around 80–100 K), involving the implementation of a cryocooler inducing significant restrictions in terms of weight, compactness and energy autonomy. Taking into account these constraints is essential to generate a new class of applications using high performance handheld thermal imagers in embedded systems, for future civil and defense applications.

Consequently, improving the temperature operation, without damaging the performance of the detectors, is currently one of the main challenges investigated by the cooled IR detector community in order to satisfy Size, Weight and Power (SWaP) criteria [1,2].

The InSb (Indium Antimonide) and MCT (Mercury Cadmium Telluride) photodetectors are the leading technologies in the MWIR domain where the presence of a strong $CO_2$ absorption line at 4.25 μm splits the MWIR window into two spectral domains usually called MWIR blue-band and MWIR red-band. The commercial InSb FPAs operate at 80–90 K in the full MWIR spectral domain with a cut-off wavelength at 5.4 μm [3] while MCT FPAs can reach operation temperatures up to 120 and 150 K with 5 and 4.2 μm cut-off wavelengths, respectively [4].

At the end of 2000 s, InAsSb XBn photodetector structures were proposed [5,6] and impressive results were obtained allowing typical operation temperatures as high as 150 K with dark current density as low as $3 \times 10^{-7}$ A/cm$^2$ [7]. With a cutoff wavelength around 4.2 μm at 150 K, such an IR system currently commercially available [8], covers only the MWIR blue-band.

One of the main advantages of Sb-based type-II superlattice (T2SL) structures is the possibility to adjust the bandgap by tailoring the layer thicknesses and the period composition, while also ensuring the lattice matching with a GaSb substrate. To extend the cut-off wavelength up to 5 μm, one can consider an InAs/GaSb T2SL [9,10]. Even if this new technology begins to be commercially available [11], Ga-containing T2SL devices suffer from a low minority carrier lifetime (100 ns in the MWIR) due to the presence of Ga-related native defects [12]. As a consequence, such T2SL detectors exhibit temperature operation lower than 110 K for a 5 μm cut-off [13].

An alternative to this technology could be the Ga-free InAs/InAs$_{1-x}$Sb$_x$ T2SL structures [14]. Indeed, an impressive minority carrier lifetime value higher than 3 μs at 80 K in the MWIR domain has been measured [15] and results on Ga-free T2SL detectors have recently been reported by research groups [16–23]. Although this new kind of detector technology operating in the full MWIR domain has recently reached significant performances, it still requires improvements in terms of dark current density values, turn on voltage, quantum efficiency and operation temperature.

In order to complete results previously reported [24], this paper describes the structural, optical and electrical measurements allowing the assess of the performance of the MWIR Ga-free T2SL detector, fabricated by molecular beam epitaxy (MBE) on a GaSb substrate.

## 2. Materials and Methods

In this section, fundamental concepts dealing both with molecular beam epitaxy (MBE) of a T2SL structure on a GaSb substrate and design of MWIR Ga-free T2SL structure and XBn unipolar barrier detector are detailed.

First of all, a choice in terms of superlattice (SL) period (*p*) and antimony composition (x) has to be made to address MWIR broadband domain but also to optimize absorption coefficient (proportional to wave functions overlap). InAs/InAsSb SL can be strained balanced on GaSb by setting the average lattice parameter of one period of the SL equal to the lattice parameter of GaSb. Consequently, InAsSb and InAs layer thicknesses ($t_{InAsSb}$ and $t_{InAs}$) as functions of the antimony composition (x) and SL period (*p*) can be calculated by using Equations (1) and (2):

$$t_{InAsSb} = ((a_{GaSb} - a_{InAs})/(a_{InSb} - a_{InAs})) \times (p/x) \tag{1}$$

$$t_{InAs} + t_{InAsSb} = p \tag{2}$$

where $a_{GaSb}$ = 6.0954 A; $a_{InAs}$ = 6.0584 A and $a_{InSb}$ = 6.4794 A stands for the lattice parameters of the binary compounds.

The MBE growth conditions of strained balanced SL structure were studied considering dedicated samples (Figure 1a). Such samples consist of a 3 μm thick InAs/InAsSb SL layers sandwiched between two GaSb layers that provide carriers barrier confinement. To assess optimal T2SL growth conditions, structural and optical measurements were performed. Structural characterizations were

made of high-resolution x-ray diffraction (HRXRD) scans and atomic force microscopy (AFM) was used to investigate the surface morphology. To perform optical characterizations, the samples were placed in a cryostat that allows a precise control of the temperature from 10 K to 300 K. To obtain photoluminescence (PL) spectra and then to evaluate the band gap energy of the T2SL structure, the samples were optically excited at 50 W/cm$^2$ with a 784 nm laser diode modulated at 133 kHz. The luminescence signal was analyzed with a Nexus 870 FT-IR system equipped with an MCT detector (12 µm cut-off wavelength). The whole optical path was under atmospheric conditions. In addition, minority carrier lifetime for InAs/InAsSb SL was extracted from time resolved photoluminescence (TRPL) measurements. For such measurements, the samples were excited by a 1.55 µm laser pulse (10 ns) at a repetition rate of 149.8 kHz, used to generate excess carriers. The power of the laser was tunable, and measurements were done at different laser pulse fluences. The photoluminescence signal was detected with a fast HgCdTe photodiode (4 ns temporal resolution, cut-off wavelength 8 µm) and analyzed with a Yokogawa oscilloscope.

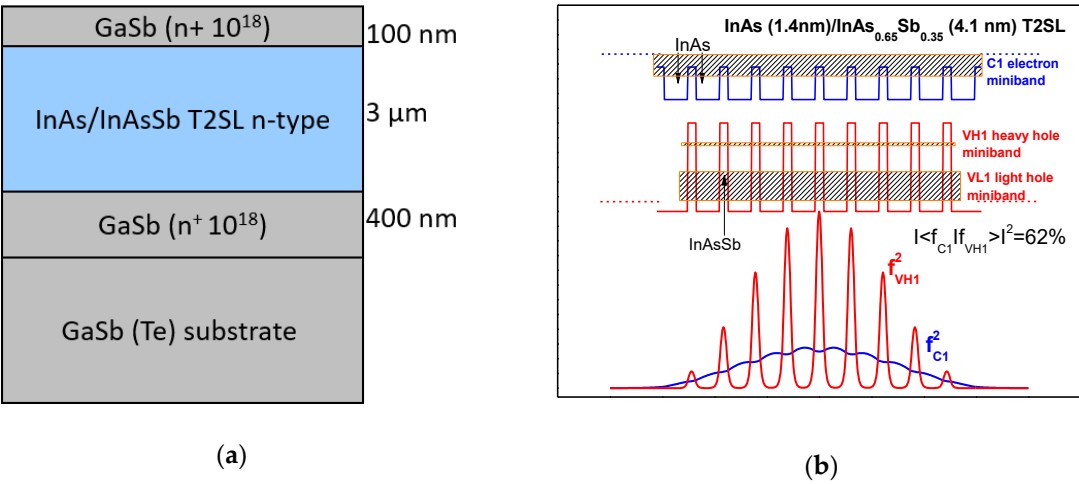

**Figure 1.** (**a**) Schematic cross section of the type-II superlattice (T2SL) structure dedicated to structural and optical measurements. (**b**) Schematic band diagram and first electron and hole minibands of Ga-free (**a**) T2SL structures. On the lower part, the fundamental electron (C1) and heavy hole (VH1) presence probability densities are reported.

With a type II-b InAs/InAs$_{1-x}$Sb$_x$ band offset [25] where electrons are confined in the binary layer (InAs) while holes are confined in the alloy one (InAsSb), the quantized miniband energies of the strain balanced InAs/InAs$_{1-x}$Sb$_x$ T2SL have been calculated with nextnano$^3$ commercial software [17]. From these data, it appears that x = 0.35 and 5 ≤ $p$ (nm) ≤ 6 are required to reach, at 150 K, a cut-off wavelength ($\lambda_{co}$) around 5 µm together with a wave function overlap between 55% and 66% (Figure 1b).

An important feature of the Ga-free T2SL material structure is the possibility of implementing it in a nBn [18] or pBn [21] MWIR barrier detector structure as absorbing layer (AL) associated with a high band gap barrier material. The main objective of the barrier structure is to reduce the contribution of the Shockley-Read-Hall (SRH) recombination current, thus the generation–recombination (GR) current to the dark current of the detector. For that, it is required to control the electric field zone by confining it in the barrier material instead of in the absorbing zone structure. Therefore, SRH processes occur in the high band gap material, instead of the IR absorbing layer. In addition, the barrier layer (BL) plays a similar role as the space charge zone in the pn structure by blocking the majority carriers and allowing the transfer of the minority ones. Consequently, this kind of device is called a bariode. When the bariode is correctly designed, GR dark current is eliminated and the dark current is diffusion

current ($J_{diff}$) limited whatever the operation temperature, improving the performances of the detector compared to a pn junction photodiode [5,6]. The diffusion current density is given by Equation (3):

$$J_{diff}(T) \propto q \times (n_i{}^2/N_d) \times (L/\tau_{diff}) \tag{3}$$

where q stands for the charge of the electron, $n_i$ for the intrinsic carrier density, $N_d$ for the doping level of the T2SL AL, L for the thickness of the AL and $\tau_{diff}$ for the minority carrier lifetime.

An $AlAs_{0.09}Sb_{0.91}$ alloy with a lattice parameter equal to the one of the GaSb substrate (Figure 2), may a priori be considered as a good candidate for barrier layer material when combined with a Ga-free InAs/InAsSb T2SL absorbing layer. Indeed, due to the intrinsic n-type doping of the T2SL absorber, holes are the minority carriers and AlAsSb barrier layer will block majority carriers, meaning the electrons.

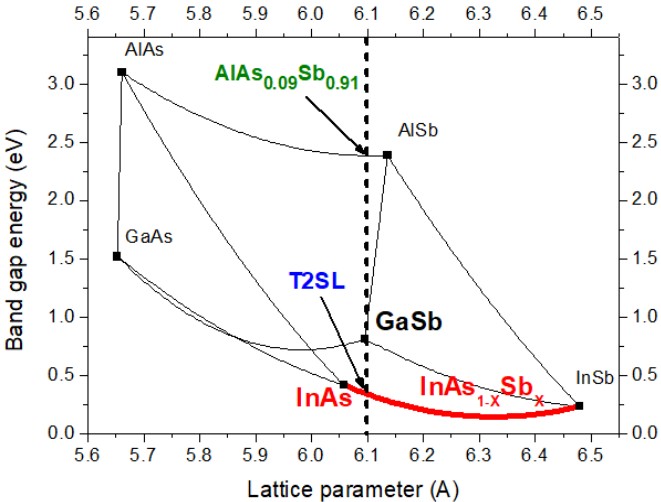

**Figure 2.** Bandgap energy vs. lattice constant for some III-V semiconductors showing the InAs/InAsSb T2SL structure and the AlAsSb ternary alloy lattice-matched to the GaSb substrate.

Figure 3a shows the stacking of the Ga-free T2SL nBn unipolar barrier photodetector. From bottom to top, the structure consists of a 400 nm Te-doped (n-type) GaSb buffer layer, which is followed by a 100 nm thick n-type doped $InAs/InAs_{0.65}Sb_{0.35}$ T2SL and by non-intentionally doped (nid) 3 µm thick absorption layer (AL) made of the same T2SL structure. A barrier layer (BL) is then made from 100 nm nid $AlAs_{0.09}Sb_{0.91}$ and finally, the contact layer (CL) of the structure is composed of an 80 nm thick n-type doped T2SL. The AL and BL are undoped, and the residual dopings are expected to be n-type and *p*-type at $10^{16}$ cm$^{-3}$ and $5.10^{16}$ cm$^{-3}$, respectively.

Figure 3b, displaying the simulated band diagram of the Ga-free T2SL nBn unipolar barrier detector at T = 150 K and V = 0 V, reveals both an accumulation layer in the AL (due to the polarities of AL and BL) and a low valence band offset between AL and BL which should not impede the transit of holes from the AL to the CL.

Such a detector structure was studied by electrical and electro-optical measurements. The T2SL devices were placed in a probe station in order to perform capacitance-voltage (C-V) measurements at a frequency f = 1 MHz and dark current density-voltage (J-V) measurements (under 0 degree field of view) for different operating temperatures. For that, a KEITHLEY 6517A Electrometer was used to both apply the bias voltage and read the current density (ratio of current and area of the device) delivered by the device. In addition, the samples were wire bounded onto a pin LCC, placed in a LN2-cooled cryostat and the non-calibrated spectral photoresponse (PR) of the detector was measured using a FTIR spectrometer.

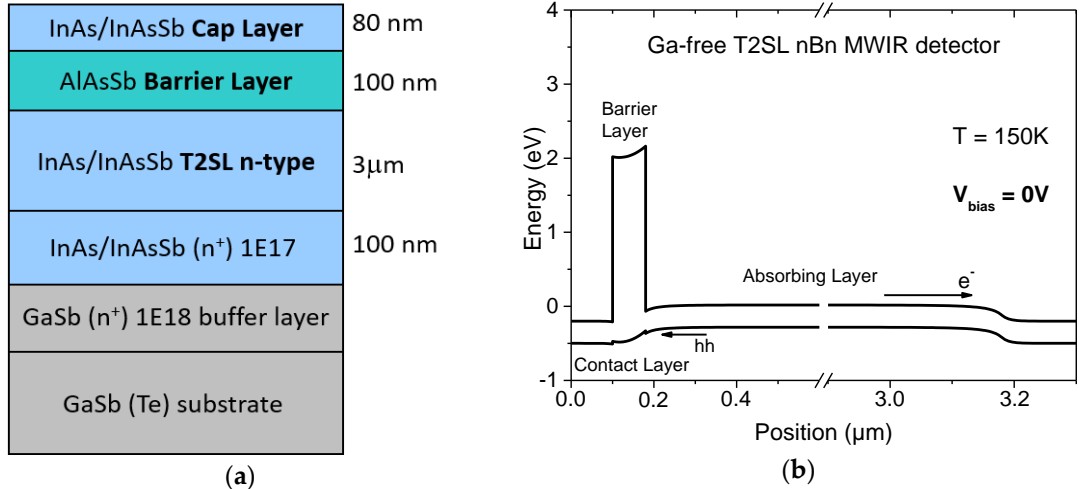

**(a)**  **(b)**

**Figure 3.** (**a**) Schematic diagram of the nBn device structure dedicated to electrical measurements. (**b**) Calculated energy band diagrams of the Ga-free nBn barrier detector structure at 150 K and 0 V.

## 3. Results

### 3.1. Fabrication and Characterizations of Ga-Free T2SL Structure

In this work, all samples were grown on an n-type GaSb Te-doped (100) substrate by solid source MBE equipped with valve crackers set up to produce $As_2$ and $Sb_2$ species. The quality of the structures grown were evaluated in terms of HRXRD, AFM, PL and TRPL measurements.

An example of HRXRD spectrum of such a T2SL structure, with $p$ = 5.3 nm, x = 34.5%, is shown in Figure 4. The presence of numerous and intense satellite peaks (SL ± 1, ±2, ±3) is a signature of the crystallographic structure's quality. Their angular separation allows the calculation of the period thickness of the T2SL while the antimony composition x of the $InAs/InAs_{1-x}Sb_x$ T2SL structure together with the lattice mismatch of the structure with the GaSb substrate, which are extracted from 0th order peak-substrate angular separation $\Delta\theta$ through Equation (4):

$$\Delta a/a = (\sin(\theta_{substrate})/\sin(\theta_{substrate} + \Delta\theta)) - 1 \tag{4}$$

where $\theta_{substrate}$ stands for the angle in degrees of the substrate peak measured by HRXRD.

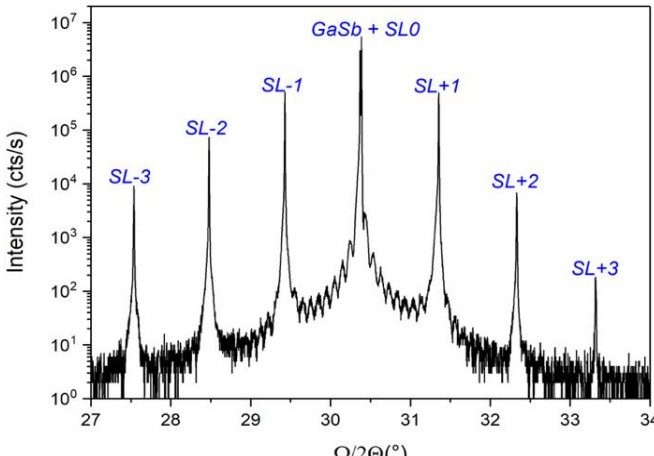

**Figure 4.** High-resolution X-ray diffraction HRXRD pattern of a $p$ = 5.3 nm Ga-free $InAs/InAs_{0.655}Sb_{0.345}$ T2SL structure grown on a GaSb substrate.

In order to check whether the epilayer has achieved a good lattice match with the substrate, typical $\Delta a/a \leq 0.1\%$ was required for the samples.

The surface morphology was also routinely investigated by AFM root-mean-square (RMS) surface roughness measurements. Typical $10 \times 10\ \mu m^2$ scan area highlights well-defined atomic steps (Figure 5) associated with RMS surface roughness only equals to 0.12 nm, that is to say, less than one monolayer in the case of Sb-based materials.

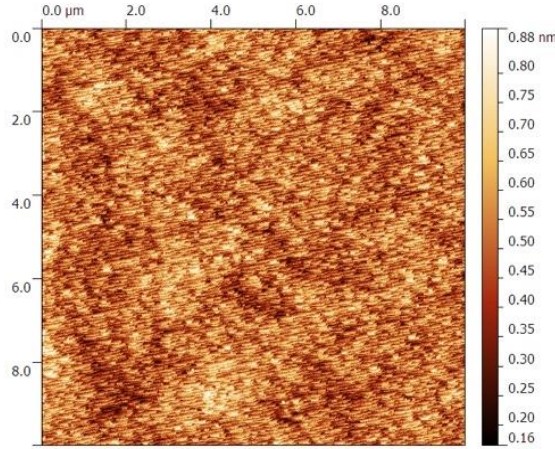

**Figure 5.** $10 \times 10\ mm^2$ atomic force microscopy (AFM) scan of an InAs/InAsSb T2SL sample. Clear monolayer steps can be observed.

PL measurements were performed from 50 to 250 K. PL spectra, presented in Figure 6, display a shift in PL peak from 4.9 $\mu m$ to 5 $\mu m$ in the temperature range (77–150 K). Such a trend strengthens the choice of the InAs/InAsSb T2SL period and antimony composition suitable for addressing MWIR broadband domain. At 77 K, the full width at half maximum (FWHM) is equal to 36 meV, higher than the one reported by E. H. Steenbergen et al. at around 30 meV [26].

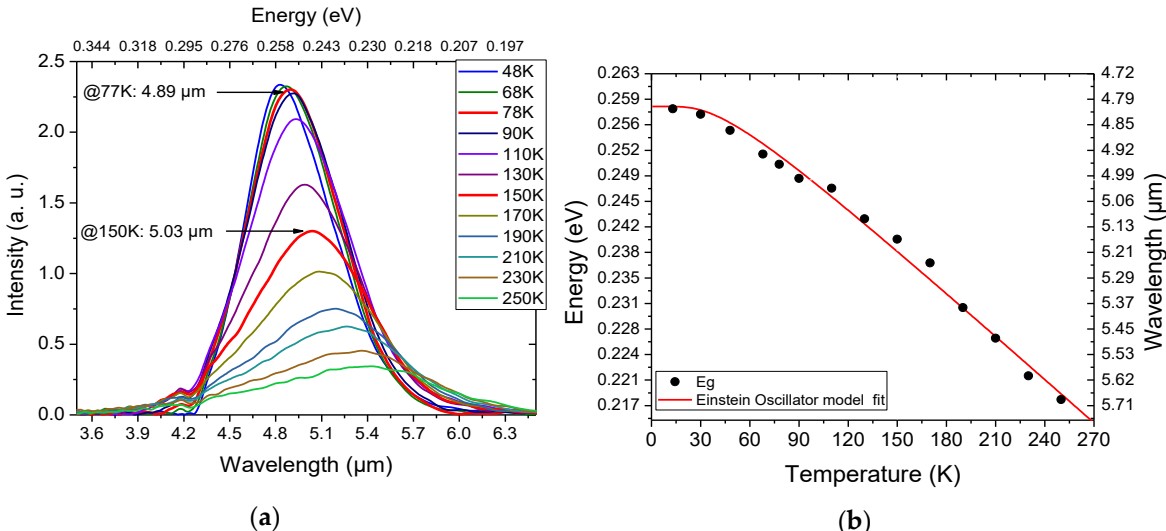

(**a**)                  (**b**)

**Figure 6.** (**a**) Photoluminescence (PL) spectra of the Ga-free InAs/InAs$_{0.63}$Sb$_{0.37}$ T2SL structure between 48 and 250 K. (**b**) Ga-free T2SL bandgap variation as a function of the temperature. The red line presents a fit curve using Einstein oscillator equation.

Using data from Figure 6a, the bandgap energy Eg(T), given by Equation (5), was plotted as a function of temperature (Figure 6b).

$$E_g(T) = E_{peak}(T) - K_B T/2 \tag{5}$$

The continuous red curve in Figure 6b represents the Einstein Oscillator model used by Webster et al. [27] and given by the Equation (6).

$$E_g(T) = E_0 - \alpha \times (T_E/(\exp(T_E/T) - 1)) \tag{6}$$

where $\alpha$ is the slope of high temperature linear asymptote, $E_0$ the energy gap at 0 K, $T_E$ the Einstein temperature and T the absolute temperature. Computing the best fit from Equation (5), values of $E_0$ = 258 meV, $\alpha$ = 2.013 × 10$^{-4}$ meV/K and $T_E$ = 117.6 K were extracted. Such fitting of the bandgap as a function of temperature will be useful to analyze dark current density measurements performed on devices [28].

TRPL measurements were performed on T2SL structures to extract the minority carrier lifetime. A typical TRPL signal measured at 90 K is reported in Figure 7. Following the approach of Donetsky et al. [29], a lifetime value as high as 1.1 μs was extracted from these measurements, a decade higher than values related to InAs/GaSb MWIR T2SLs [12,29,30]. Such a lifetime value validates both the MBE growth procedure and the choice of Ga-free T2SL structure as AL for the device. However, because heavy holes are mainly confined in the InAsSb layer, vertical transport will have to be investigated. Very recent results show that heavy hole mobility is strongly temperature dependant [31,32].

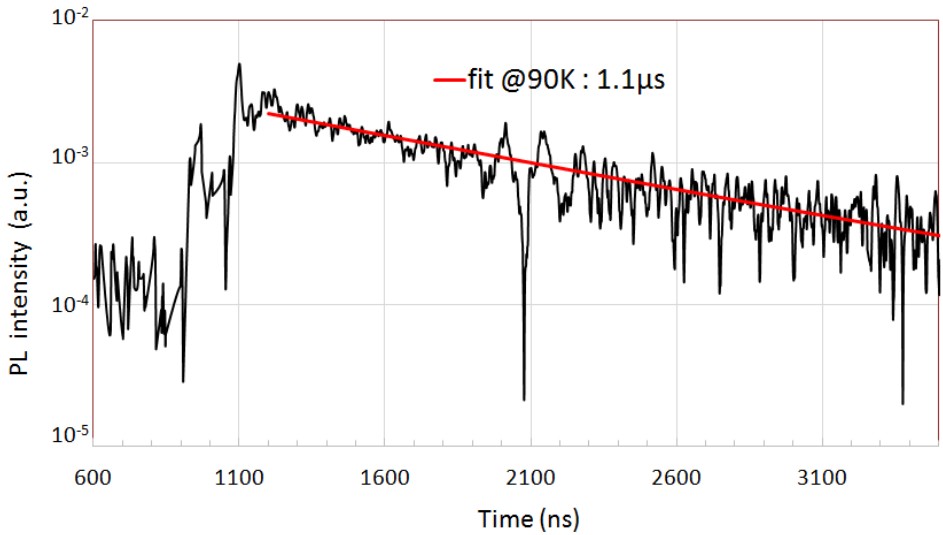

**Figure 7.** TRPL signal of the Ga-free InAs/InAs$_{0.63}$Sb$_{0.37}$ T2SL structure at 90 K. A minority carrier lifetime of 1.1 μs is extracted.

The Table 1 summarizes the structural and optical baseline values routinely used as a quality indicator to evaluate the performance of Ga-free InAs/InAsSb T2SL structures.

**Table 1.** Structural and optical criteria used to evaluate Ga-free T2SL structures grown by MBE.

| HRXRD Δa/a | AFM RMS | PL λ$_{peak}$ @ 150 K | PL FWHM | TRPL Lifetime |
|---|---|---|---|---|
| ≤0.1% | ≤0.15 nm | 5 μm | ≤0.30 meV | ≥800 ns |

### 3.2. Fabrication and Characterizations of Ga-Free T2SL Barrier Detector

From epitaxial T2SL structures, circular mesa nBn photodetectors with diameters from 60 μm to 310 μm (Figure 8) were fabricated using standard photolithography. Mesa photodetecors were realized by wet etching using citric acid solution and polymerized photoresist was used to protect the mesa surface from ambient air. Metallization was ensured on the n-GaSb substrate and on the n-type T2SL cap layer. The barrier detector devices were then characterized by electrical and electro-optical measurements.

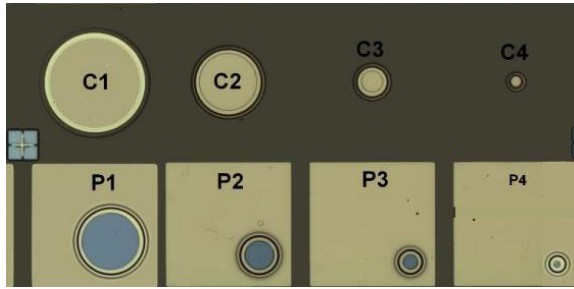

**Figure 8.** Top view of a processed sample, blind diodes (C) and photodetectors (*p*) with several diameters from 60 μm up to 310 μm.

Figure 9 shows non-calibrated PR spectra recorded at different bias (from −60 mV to −1.50 V) and at 150 K. The bias-dependent PR signal increases with increasing reverse bias and starts to saturate at −220 mV. In addition, the PL peak at 5.03 μm is in good agreement with the 50% cut-off wavelength ($\lambda_{co}$) extracted from the spectral PR.

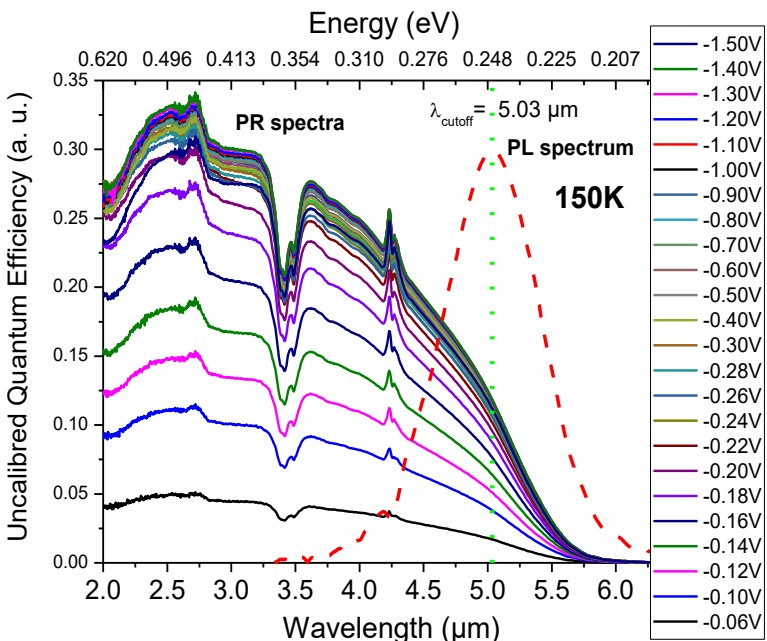

**Figure 9.** Photoresponse (PR) at different biases and photoluminescence (PL) measurements at 150 K.

Dark current density was measured and Figure 10 shows the dark current density-voltage (J-V) characteristics carried out for a 210 μm diameter detector at different temperatures, from 77 to 270 K. At the expected temperature operation of 150 K and bias equal to −395 mV, the dark current density is equal to $3.24 \times 10^{-5}$ A/cm². Below 110 K, the dark current is limited by the photonic current due to the

experimental set-up. The shapes of the J-V characteristics are in accordance with those related recently reported XBn Ga-free MWIR diodes [16–23].

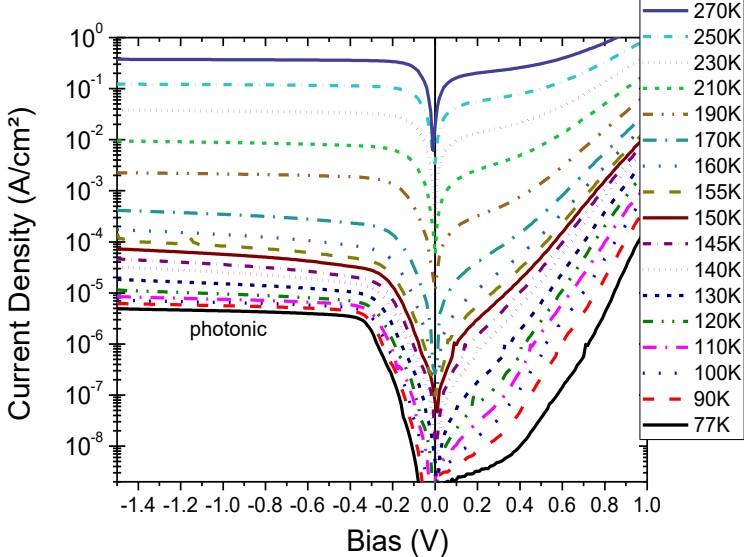

**Figure 10.** Dark current density-voltage (J-V) characteristics of a Ga-free T2SL nBn MWIR detector for temperatures from 77 to 270 K.

To complete electrical characterizations, C-V measurements were performed on the devices. From these measurements, typical $1/C^2$ data as a function of voltage obtained at T = 150 K are shown in Figure 11. Using the Equation (7) [33], the extracted slopes allow to reaching of the residual carrier concentration ($N_{res}$) both in the nid *p*-type BL ($5.2 \times 10^{15}$ cm$^{-3}$) and in the nid n-type AL ($3.2 \times 10^{15}$ cm$^{-3}$) of the nBn device.

$$(A/C)^2 = (2/(q\varepsilon_0\varepsilon_{SL})) \times ((V_d - V)/N_{res}) \tag{7}$$

where A stands for the active area of the photodetector, $V_d$ for the diffusion potential, q for the charge carrier, $\varepsilon_0$ for the vacuum permittivity and $\varepsilon_{SL}$ for the relative permittivity of the InAs/InAsSb T2SL ($\varepsilon_{SL}$ = 15.15).

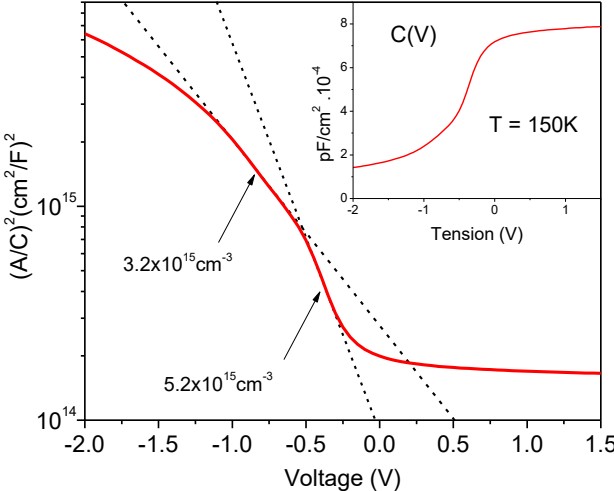

**Figure 11.** Capacitance-voltage measurement of a Ga-free T2SL nBn detector at T = 150 K. $1/C^2$ (V) characteristic and in inset, surface capacitance $C_{surf}$ (V) curve.

## 4. Discussion

Several analyses can be made concerning the obtained results. To explain the behavior of the PR spectrum as a function of the bias applied (Figure 9), a band diagram of the nBn device was considered. In addition to the band diagram calculated at 150 K and V = 0 V (Figure 3b), Figure 12 reports two more band diagrams calculated at −100 mV and −500 mV. At V = 0 V, the band diagram highlights the presence of a potential barrier blocking the minority heavy hole carriers. Even, at a bias operation equal to −100 mV, this potential barrier remains, penalizing the quantum efficiency. At V = −500 mV, the bias is high enough to suppress the potential barrier allowing the transport of a hole minority carrier to the top contact layer. The maximum value of the photo-generated carriers is then collected. The vanishing of the potential barrier starts at V = −220 mV, explaining the saturation of the PR signal recorded. This analysis shows that the choice of the AlAsSb BL and the T2SL CL with their corresponding dopings is probably not optimal.

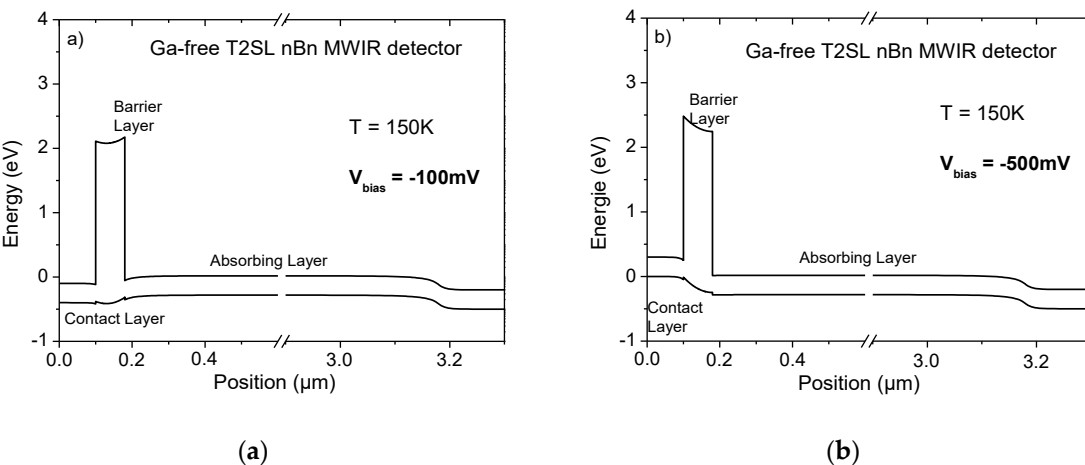

**Figure 12.** Calculated band diagrams of the Ga-free T2SL nBn barrier structure at 150 K and (**a**) −100 mV, (**b**) −500 mV.

An arrhenius's plot can be extracted from J-V characteristic curves (Figure 10). Such a graph (Figure 13) shows an activation energy Ea = 225 meV at a high temperature, which is approximately the energy of T2SL bandgap. Such a value is in accordance with a diffusion current (Equation (3)) regime. At temperatures lower than 140 K, the device is GR dark current density limited, evidencing the presence of an unwanted electric field in the T2SL AL. The presence of the electric field in the AL may be suppressed by a better control of doping levels during the MBE growth, both for BL and AL. Moreover, a dark current value equal to $3.24 \times 10^{-5}$ A/cm$^2$ at 150 K (Figure 10) has to be improved. Indeed, such a result, compared to the MCT state of the art of photodiode limited by diffusion dark current [34], is 20 times higher at the corresponding cut-off wavelength and remains slightly superior to the most recent results reported on Ga-free T2SL detectors [18–23]. To lower the dark current, the $N_d$ x $_{diff}$ product must be as high as possible (Equation (3)). A complementary study to determine the optimal Nd x $\tau_{diff}$ product is necessary and planned.

Electrical results and spectral PR at 150 K may be jointly and closely analyzed in order to define the operating bias and to explain the different dark current regimes as a function of bias. For this purpose, Figure 14a–c show the normalized PR, the dark current density and the $R_dA$ product, respectively. $R_d$ represents the differential resistance, calculated from the derivative of the voltage over the current and A is the device area. The normalized PR values were extracted from the spectral photoresponse measurements at different biases and at 4 μm (Figure 9).

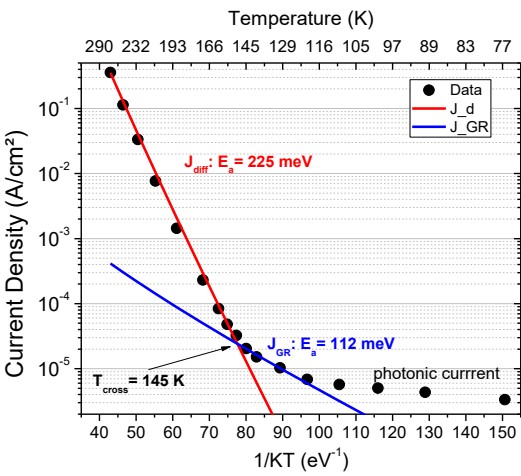

**Figure 13.** Arrhenius's plot extracting from J-V curves at −350 mV.

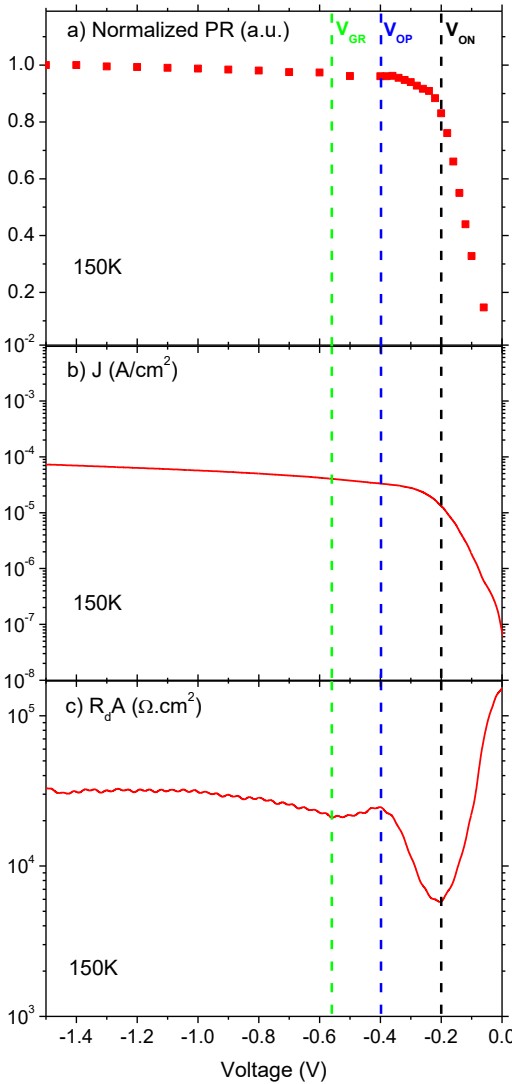

**Figure 14.** Stacking of the experimental (**a**) normalized photoresponse, (**b**) dark current density, (**c**) Differential Resistance Area product (RdA) product as a function of the voltage at 150 K for the Ga-free T2SL MWIR nBn photodetector.

By examining the shape of the displayed curves at 150 K, we can identify three main dark current regimes. The first significant bias value, $V_{on}$ (turn on bias), is located at the first $R_dA$ minimum at $-200$ mV. At this bias, thermal-generated holes can reach the CL and the normalized PR is higher than 90%. Next, the $R_dA$ peak at $-395$ mV, equal to $2.5 \times 10^4$ $\Omega$ cm$^2$, corresponds to the operating bias $V_{op}$ (operating bias). The photoresponse is maximized and the device is fully turned on at this bias. As a consequence, it is not necessary to increase the bias among $V_{op}$ when the photodetector is operating. The depletion region is still confined within the barrier until $V_{op}$ and the C-V measurement permits determination of the reduced carrier concentration in the BL (Figure 11). Then, the next $R_dA$ minimum at $-580$ mV indicates that the GR current starts to appear. At this bias, the barrier is fully depleted and the depletion region reaches the absorber, which explains the appearance of the electric field-related GR current. We define $V_{GR} = -580$ mV and at higher bias, the reduced carrier concentration in the AL is determined with C-V measurement (Figure 11).

Results obtained show that the observed operating bias value ($-395$ mV) is higher than the expected one to satisfy SWAP criteria. This drawback could be due to a misalignment between the AL and BL valence bands, impeding the flow of minority carriers. To circumvent it, future design improvement is required.

## 5. Conclusions

In conclusion, we have reported a set of structural, optical and electrical characterizations performed on MWIR Ga-free InAs/InAsSb T2SL barrier quantum detectors grown by MBE on a GaSb substrate in order to evaluate their performances and the way to improve them. The photodetector under study showed a cut-off wavelength around 5.03 µm at 150 K. The dark current measurements were then analyzed to identify different current mechanisms in the structure and to estimate the operating bias. The operating bias ($-380$ mV) was higher than expected for SWAP conditions and this issue could be reduced by both adjusting the valence band alignment between the absorber and barrier layers and by optimizing their residual doping levels during the MBE growth. At this operating bias, a dark current density equal to $3.24 \times 10^{-5}$ A/cm$^2$ was extracted from J(V) measurement at 150 K. Compared to the dark current state of the art of photodetector limited by diffusion current, this value has to be lowered by optimizing the $N_d$ x $\tau_{diff}$ product. It will be the main subject of forthcoming studies.

**Author Contributions:** U.Z.-M., M.B., R.A. and J.P.P. fabricated the structures and devices; U.Z.-M., M.B., R.A., and S.B. performed the measurements; U.Z.-M., M.B., J.-P.P., I.R.-M., F.d.A.-S. and P.C. analyzed the data; U.Z.-M., J.P.P., F.d.A.-S., and P.C. wrote the paper. All authors have read and agreed to the published version of the manuscript.

**Funding:** This work was partially funded by the French "Investment for the Future" program (EquipEx EXTRA, ANR 11-EQPX-0016) and by the French ANR under project HOT-MWIR (ANR-18-CE24-0019-01).

**Conflicts of Interest:** The authors declare no conflict of interest.

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
