# Peer review of "Structural, Optical and Electrical Characterizations of Midwave Infrared Ga-Free Type-II InAs/InAsSb Superlattice Barrier Photodetector"

_photonics, doi:10.3390/photonics7030076_

Round 1

Reviewer 1 Report

In this manuscript, the authors fabricate a Ga-free InAs/InAsSb type-II superlattice, grown by MBE on GaSb substrate. In my opinion, this manuscript is interesting to the readers of photonics. The topic is very important in this field. This work is novel and original. The authors have solid background in this field. Therefore, the referee recommends it to be published after the following minor revisions:
1.  The English should be polished by a native speaker.
2.  It is suggested the authors to check their manuscript carefully and thoroughly to avoid some typical mistakes. For example,

line 16, 1 μs (not 1μs), line 16, 80 K (not 80K), Wavelength (not wavelength) in Fig. 6(a)

3. How are the performances here compared with state-of-the-art reports for IR photodetectors? The readers would like to see a paragraph near the end of the manuscript before the conclusion to dedicate to such comparison.

4. More recent literature for photodetectors are suggested to be included in the introduction section, e.g. Nature Communications. 10 2125 (2019), Journal of Non-Crystalline Solids. 546 120292 (2020), Journal of Alloys and compound 804 5 (2019), and Sensors 18(11), 3755 (2018).

In general, this work seems to be very interesting. The referee would like to see the revision if possible.

Author Response

Dear Editor,

We would like to thank the reviewer for its valuable and constructive comments on our manuscript. We have modified the original manuscript according to its comments, including English polishing. All changed parts of the text are highlighted in yellow in the new version.

Answer to reviewer 1 comments :

- The English should be polished by a native speaker.

→ The paper has been carefully re-read and re-written. In the new version, changes made have been highlighted in yellow.

- It is suggested the authors to check their manuscript carefully and thoroughly to avoid some typical mistakes.

→ In addition of the previous comment, many typographic errors have been corrected, especially dealing with physical quantities and their units.

- How are the performances here compared with state-of-the-art reports for IR photodetectors? The readers would like to see a paragraph near the end of the manuscript before the conclusion to dedicate to such comparison.  

→ We would like to thank the reviewer for this comment. Before the conclusion (end of page 10), a comparison with state of the art has been re-written and completed.

- More recent literature for photodetectors are suggested to be included in the introduction section, e.g. Nature Communications. 10 2125 (2019), Journal of Non-Crystalline Solids. 546 120292 (2020), Journal of Alloys and compound 804 5 (2019), and Sensors 18(11), 3755 (2018).

→ We have chosen not to take into account this suggestion made by the reviewer for purposes of our study and justified this statement in the following. Indeed, our paper is focused on antimonide-based III-V semiconductor interband T2SL quantum detector operating in the midwave infrared spectral domain (3-5µm) and the proposal additional papers are dedicated to intraband QDIP structure, UV and near IR (1.55µm) detectors. We think that these works are out of the scope of the paper. In the introduction, the most recent and pertinent literature on the subject is given by the papers 16 to 23 of the references list. However, we have still updated the bibliography and added references 31 and 32 on vertical transport in Ga-free T2SL structure.

  • - Tsai, C-Y.; Zhang, Y.; Ju, Z.; Zhang, Y-H. Study of vertical hole transport in InAs/InAsSb type-II superlattices by steady-state and time-resolved photoluminescence spectroscopy. Phys. Lett. 2020, 116, 201108-201108-5
  • - Casias, L.K.; Morath, C.P.; Steenbergen, E.H.; Umana-Membreno, G.A.; Webster, P.T.; Logan, J.V.; Kim, J.K.; Balakrishnan, G.; Faraone, L.; Krishna, S. Vertical carrier transport in strain-balanced InAs/InAsSb type-II superlattice material. Phys. Lett. 2020, 116, 182109-182109-5

Reviewer 2 Report

Zavala-Moran et al. presents an in-depth investigation of the structural, optical and electrical properties of midwave infrared detectors. The authors simulate and fabricate two types of barrier detectors based on InAs/InAsSb type II superlattice on GaAs substrate. The authors' main objectives are to extend the cut-off wavelength of these detectors into the MWIR red-band, while maintaining low dark currents and increasing operating temperatures.

The study carried out is quite thorough. The motivation and state-of-the-art are clearly exposed. The presentation of the design/characterization results is well-structured and comprehensive. The discussion also clearly stresses the paths for further improvements/understanding. All in all, the obtained results are convincing and hold promises for improving the performances of this type of detectors. I would thus recommend publication of this work in Photonics.

The authors may want to check their manuscript for typos and the english formulation in places (e.g. l. 229: "is as low", l. 238 "thanks to equation 7").

Author Response

We would like to thank the reviewer for his comment on our manuscript. As suggested, we have carafully checked the manuscript to avoid English mistakes and typographic errors.

All changed parts of the text are highlighted in yellow in the new version.

Reviewer 3 Report

The manuscript under review with Photonics journal, entitled “Structural, optical and electrical characterizations of midwave infrared Ga-free type-II InAs/InAsSb superlattice barrier photodetector” reports on a comprehensive experimental study of mid-IR barrier photodetectors. This includes a detailed testings of both structural and opto-electrical characterizations. The paper built upon a previous work and results reported in Ref. [24].

The paper itself is well-organized and well-written, and thus easy-to-follow for readers. The figures are clear. Provided results, their presentation/interpretation as well as overall discussion is solid. Authors reported on an extensive set of measurements and results, which can be interesting for the community working in this particular field.

In my opinion, the work reported in the manuscript can be considered for publication. Some recommendations and questions for the manuscript are given below.

(1) Page 1, line 4. Please remove the dot in the title.

(2) Be consistent when using abbreviation in the manuscript. Some of them are correctly written such as “molecular beam epitaxy (MBE)”, but several are like this “InSb (Indium Antimode)”.

(3) Moreover, please make it clear for readers what is “the dark current” and what is “the dark current density”. On many places in the manuscript, those terms are wrongly used, which can be misleading for readers. Please report in the manuscript the measured values of the dark current. Now, only dark-current density is provided. Also provide definition of the dark-current density.

(4) Why did Authors perform only dark current tests and not the test for the device photo-current? It seems like that the dark-current for the device is quite low, so I am wondering about the level of the photo-current that could be generated by the device. This why additional important device characteristics can be provided for the readers such as device responsivity, effective device on-off ratio (photo-current by dark-current), device linearity, detector spectral reach or even device gain?

(5)Can Authors comment on the device speed or radio-frequency response?

(6) What about the target applications for this kind of detectors?

Round 2

Reviewer 3 Report

No more comments on this manuscript.